# Leveraging Grief: Involving Bereaved Parents in Pediatric Palliative Oncology Program Planning and Development

**DOI:** 10.3390/children8060472

**Published:** 2021-06-03

**Authors:** Holly L. Spraker-Perlman, Taylor Aglio, Erica C. Kaye, Deena Levine, Brittany Barnett, Kathryn Berry Carter, Michael McNeil, Lisa Clark, Justin N. Baker

**Affiliations:** 1Division of Quality of Life and Palliative Care, Department of Oncology, St. Jude Children’s Research Hospital, Memphis, TN 38105, USA; holly.spraker-perlman@stjude.org (H.L.S.-P.); erica.kaye@stjude.org (E.C.K.); deena.levine@stjude.org (D.L.); lisa.clark@stjude.org (L.C.); 2Department of Global Medicine, St. Jude Children’s Research Hospital, Memphis, TN 38105, USA; taylor.aglio@stjude.org (T.A.); michael.mcneil@stjude.org (M.M.); 3Family, Guest and Volunteer Services Program, St. Jude Children’s Research Hospital, Memphis, TN 38105, USA; brittany.barnett@stjude.org (B.B.); kathryn.berry-carter@stjude.org (K.B.C.)

**Keywords:** pediatric palliative care, bereavement care, bereaved parents, pediatric oncology bereavement

## Abstract

As pediatric palliative care (PPC) became a recognized medical specialty, our developing clinical PPC team longitudinally partnered with bereaved parents to understand the care that their children received as they transitioned towards end of life. Families developed Eight Priorities, shared within, to improve care for children with a poor chance of survival based on their experience of losing a child to cancer. In this paper, we delineate the top eight PPC needs from a parent perspective to offer multi-layered, individually tailored resources for patients and families. One of these Eight Priorities noted that bereavement care for the remaining family members is vital for healing after the death of a child to promote meaning making and resilience in bereaved families. Here, we outline the creation of a bereaved parent-designed bereavement support program as one example of how we have partnered with parents to fulfill their Eight Priorities for quality care.

## 1. Introduction

After the death of a child, parental bereavement is complex and associated with adverse health outcomes, including poor mental and physical health [1]. Bereaved parents have identified a need for improved bereavement support, emphasizing the important role that the healthcare team and medical institution serve in their grief journey [2]. Despite this need, care for bereaved parents did not exist widely 10 years ago, and pediatric providers might not have knowledge, even when bereavement care programs exist [3]. Formal recommendations for a standardized approach to bereavement care for families whose child dies from cancer were not published until 2015 in the Standards for the Psychosocial Care of Children with Cancer [4]. These standards emphasized the importance of maintaining meaningful contact between the healthcare team and bereaved parents and providing resources for bereavement support.

Recognizing the need for improved psychosocial and emotional support for patients with limited life expectancy and their families, we partnered with bereaved parents at an academic hospital focusing on the care of children with cancer and blood disorders to determine priorities for institutional palliative care and bereavement programming. To delineate the unmet concerns and needs of parents of children with cancer, we created an institutional task force that included 13 bereaved parents who used their personal experiences to draft 8 priorities for pediatric palliative oncology needs. Since adopting these 8 priorities, we have made progress towards the provision of holistic care for families of children with life-threatening illnesses. 

We have leveraged motivated bereaved parents to design, build, and staff programming for current patients and families, healthcare providers, and students that serves as a model for other institutions [5,6]. In this paper, we describe the process of empowering bereaved parents, in collaboration with staff, to honor their child’s legacy and make meaning of their child’s illness and death through their commitment to optimize quality of care for current and future families. In 2010, a subcommittee of the Patient Family Advisory Council (PFAC) was formed to address quality of life and palliative care needs at the institution. This initial subcommittee included 13 bereaved parents. Two physicians conducted a series of focus groups to better understand the care priorities of bereaved parents based on their personal experience of cancer treatment and bereavement support at our institution (Appendix A). Data from these focus groups were subsequently discussed, analyzed, summarized, and revised to reflect group consensus [7]. The identified priorities were crafted into 8 parent-derived recommendations for improvement in palliative and end-of-life care for pediatric oncology patients and are summarized below (Table 1). While our institutional mission focuses on the provision of care for children with catastrophic illness, primarily pediatric cancer and blood disorders, we feel that these parental priorities for quality care may be applied to pediatric patients with other chronic and/or serious medical conditions.

## 2. Parental Recommendations to Improve the Quality of Pediatric Palliative and End-of-Life Care (“The Eight Priorities”)


**Recommendation #1: To ensure that children receive the best possible treatment of disease and have the best possible quality of life, always hoping for the best possible outcome.**


Parents have an overwhelming desire to cure their child’s illness, when medically feasible. They also prioritize allowing their children a life as normal as possible given the circumstances. Parents want to collaborate with and be empowered by the healthcare team to make their child’s life the best that it can be each day.

The healthcare team must be empathetic to the personal needs of sick children from diagnosis to either survival or death and bereavement. Parents advocate for healthcare that is tailored to the needs of their child and that works towards patient- and family-defined goals. Healthcare providers should be prepared to support the emotional needs of parents and siblings in the context of their spiritual, personal, and/or cultural values. Thus, they must be educated in not only providing curative therapies, when available, but also in empathetically supporting families in the process of decision making when cure is not an option.


**Recommendation #2: To provide effective symptom management throughout the illness trajectory.**


Parents have a primal desire to keep their children safe and comfortable. Parents suffer when their child experiences pain or refractory symptoms while receiving medical treatment. Families want to work with healthcare providers to ensure their child’s comfort through effective management of distressing symptoms. Some families have personal, cultural, or religious beliefs that may influence their thoughts or concerns about management of symptoms, including pain.

Healthcare providers must recognize that physical comfort and quality of life are intertwined for patients and families. Effective symptom management is important throughout the illness trajectory; however, it is critical as cure becomes less likely. Families advocate that critically ill children deserve expert-level pain and symptom management, and families should be able to express their beliefs about balancing management of symptoms with disease-directed therapy.


**Recommendation #3: To provide relationship-based care.**


Parents want trusting and meaningful relationships with their child’s healthcare team, seeking compassionate and skilled healthcare staff who create opportunities for connection and support. Healthcare teams must query and address patient and family concerns, hopes, and values to build these relationships.

Healthcare providers must acknowledge pediatric patients as vital members of a larger family unit and provide care directed towards the family as a unit. When a child is diagnosed with a life-threatening illness, all family members are affected.


**Recommendation #4: To empower families with useful and reliable information.**


Parents feel empowered to make decisions on behalf of their children when given useful and reliable information. Parents must navigate an increasingly complex healthcare system and advocate for their children and can only effectively participate in their child’s care when provided pertinent, honest information. 

Healthcare providers must be properly trained to communicate productively with children and family members, particularly when the information may be difficult to receive. Healthcare providers should not delay difficult conversations as this lag may rob parents of time to have meaningful conversations with their child and family about what to expect if the cancer treatment fails. Parents want providers to support their hope while offering transparent prognostic information. 


**Recommendation #5: To support children and families in the process of making difficult care decisions.**


Parents of children with cancer recognize their obligation to make the “right decisions” on their child’s behalf and believe that end-of-life decisions should be family centered. Parents fear they will regret decisions made for their children and rely on guidance and support from healthcare providers for reassurance.

Healthcare teams provide information, options, and recommendations, but parents ultimately make preference choices that provide peace. Healthcare providers must be adequately trained to help families to establish a plan of care that is consistent with a family’s beliefs and values, and to be respectful and supportive of that decision.


**Recommendation #6: To facilitate care coordination.**


Healthcare should be well organized and streamlined, but many obstacles exist for seriously ill children. For example, many sick children are cared for by multiple healthcare teams in several locations, making coordination of appointments, medical information and plans, and collaborative management a challenge.

Healthcare providers from a variety of disciplines must communicate and collaborate effectively to properly care for children with life-threatening illnesses. For children with cancer, this may include oncologists, surgeons, radiation oncologists, physical/occupational/speech therapists, psychologists, social workers, nurses, and others. This is crucial at times of crisis, such as during periods of hospitalization or transition from the inpatient to the outpatient setting, from the hospital back to their home community, and to hospice care near the end of life. Parents need guidance and support to make care transitions and navigate the healthcare system.


**Recommendation #7: To ensure that children with progressive, incurable illness experience a comfortable and peaceful death.**


When children have progressive disease and death is expected, most parents want to ensure that their children are comfortable and free from unnecessary suffering. In the presence of advancing illness, the healthcare team should prepare parents for the possibility that their children may die, and attention should be focused on eliminating discomfort as perceived by the healthcare team and family members. When medically possible, parents should be able to choose their child’s location of death, as where a child dies can affect families for years after their loss [8,9] Healthcare providers should reduce familial distress during the end of life period by providing anticipatory recommendations for the best known management of distressful symptoms to support the best possible death. 


**Recommendation #8: To provide bereavement support for surviving family members and hospital staff.**


The death of a child is a tragic event of immeasurable consequence in the lives of surviving family members. Ideally, healthcare systems and mental health providers are available for grief and bereavement support for siblings and parents. The goal of bereavement care is to help individual family members transition back to a comfortable, productive life after the death of their loved one.

Healthcare providers need education about how to sensitively communicate with bereaved family members to mitigate feelings of abandonment that many families describe. Healthcare providers should create opportunities for bereaved siblings and parents to talk about their family member and share their experiences. They must also recognize that mothers, fathers, and siblings grieve differently and that the bereavement needs of each family member should be addressed separately. Healthcare teams are a valuable source of comfort and support for families. Parents recognize that when a child dies, medical teams who cared for that child may also feel enormous grief. Parents want healthcare providers to have access to tools that help them process their grief, too, to continue their work and avoid burnout.

## 3. Progress towards These Goals

In 2011, these recommendations were presented by a parent to institutional administration, and in alignment with the parents’ top priority (comprehensive bereavement care, which did not exist at that time), the first Bereavement Coordinator was hired in 2012. That year, the Quality of Life Steering Council (QoLSC) was also prioritized and created as a formal branch of the Patient- and Family-Centered Care (PFCC) program (Figure 1). Through this committee, a rotating group of diverse bereaved parents were provided a platform to leverage their personal grief experiences to advocate for improvement in the care of families of children with cancer. Below, we present several examples of ways in which bereaved parents have used their intimate knowledge to change the hospital’s culture, focusing on our partnership with parents in the creation of a comprehensive bereavement care program (Recommendation #8) as an exemplar of our institutional commitment to these recommendations.

## 4. Quality of Life Steering Council (QoLSC)

The QoLSC and the Patient and Family Advisory Committee (PFAC) are both subsets of the Patient- and Family-Centered Care Program (PFCCP). The PFAC is comprised of parents of current and former patients who are doing well from a cancer-directed standpoint, and healthcare providers. The QoLSC and the PFAC both contain parent members (either bereaved or not), and work together to prioritize care initiatives across the entire institution for all patients under the PFCCP. The Quality of Life Steering Council (QoLSC) was formed to specifically enable the voices of bereaved parents to be reflected in all aspects of care, from routine medical care to palliative care institutional initiatives to hospital strategic planning. The QoLSC is committed to ensuring that children receive the best possible treatment, have the best possible quality of life, are supported along with their families in making difficult care decisions, experience a comfortable and peaceful death, and that bereavement support is provided for surviving family members and staff after the death of a child. Several innovative programs have been developed by the QoLSC to improve quality of care.

The QoLSC is composed of 11 bereaved caregivers of deceased patients and 6 staff members, under leadership from the Division of Quality of Life and Palliative Care and from Family, Guest, and Volunteer Services. To participate on the QoLSC, caregivers must be at least 2 years past the death of their child. Prospective members may be nominated by staff or family members or may be self-referred. Candidates complete an application and additional input is gathered from the Patient- and Family-Centered Care staff, Bereavement Coordinator, and clinical staff to assess the applicant’s ability to effectively collaborate in a group. All members are onboarded as hospital volunteers, screened, and interviewed by various staff members, background checked, and trained for their role. Parents have varied backgrounds including gender, race, socioeconomic status, and cancer diagnoses (of their child), and they serve a term of 1 year (with up to 3 renewals before vacating their seat). Parents serving on the QoLSC meet monthly to plan educational interventions and events, review current bereavement programming, identify gaps in care, and offer input about the institution’s quality improvement initiatives [Recommendation #1]. 

## 5. Quality of Life for All (QoLA) Clinical Team

When the bereaved parental recommendations were originally developed, the institution had only very recently started a formal clinical pediatric palliative oncology consultation service. The Quality of Life for All (QoLA) team began providing palliative care consultation and follow up appointments institution-wide in 2008. After partnering with the bereaved parent focus group members and the QoLSC, planning began to expand the pediatric palliative care medical team to help address all eight parental recommendations. The partnership between the QoLA providers and these initial bereaved parents, and their shared personal healthcare experiences helped to shape the way the QoLA Clinical service developed and practices today. Initially, the QoLA service consisted of two part-time physicians and one nurse practitioner to meet quality of life needs for all patients and families at our institution and in their home communities. To meet these complex care needs, additional staff have been added over the past 13 years, and the QoLA team now consists of 4 full-time and 3 part-time board-certified pediatric palliative care physicians, 5 nurse practitioners, 1 nurse coordinator, an embedded part-time chaplain, and a psychologist who also functions as the bereavement coordinator [10]. Parents of children at our institution have been valuable advocates for an expansion in clinical palliative care programming by sharing personal narratives surrounding the gaps in care they may have experiences as their children moved towards end of life with our administration. For example, the difficulty in transporting a seriously ill child back and forth to the hospital for medical care that could be provided in the patient’s home led to expansion of a hospice service to also provide home based palliative care. The QoLA inpatient service works very closely with a home-based care team of 5 nurses, 2 social workers, a chaplain, a child-life specialist, and a team coordinator to assure that patients can spend as much time at home as possible while having access to medical services. This team provides expert-level symptom management, relationship-based care, empathetic prognostic information, assistance with medical decision making, cohesive care coordination (including home-based support), and bereavement programming for patients, families, and staff for hundreds of patients per year [Recommendations #1–8].

## 6. Bereaved Parent Educators

To better meet the needs of families, clinicians benefit from formal training in communicating difficult news. QoLA faculty members created educational initiatives in which bereaved parents model provision of prognostic information and empathetic support to promote shared medical decision making among clinicians, patients, and families [2,11] [Recommendations #4 and 5].

Under the direction of the Bereavement Coordinator and the Quality of Life Education Coordinator, bereaved parents receive formal training in how to teach communication skills. These parents offer their unique perspectives while emphasizing empathetic listening, verbal and non-verbal communication techniques, and strategies to approach sensitive conversations. The story of the bereaved parent’s journey with their child from diagnosis to bereavement becomes the educational platform for students, residents, fellows, and staff. Educational events primarily occur at our institution but may also occur at regional and national institutions and/or conferences. All pediatric fellows, of any discipline, have multiple formal opportunities to learn from the bereaved parent educators throughout their training (Table 2) [2,5,6]. 

## 7. Bereavement Program

Based on parental recommendations, a formal bereavement care program is now offered to surviving family members [Recommendation #8]. After a patient’s death, an individualized bereavement plan note is created in the patient’s medical record. The plan details the minimum level of bereavement support that is to be provided to each bereaved family. A bereavement checklist, which documents how the bereavement plan is being carried out, is also entered into the medical record. After the death of their child, families are provided at least these five supports:Mailing Program: Each family receives four bereaved parent-created mailings during the first year following the death of their child. In addition, the QoLA clinical service and psychosocial teams sign and mail condolence cards to all families. Finally, for patients enrolled in the institutional School Program, a condolence card signed by their teacher(s) is sent and a book is donated in the child’s memory to their local school.Funerary assistance: Funerary assistance is provided to families (paid directly to the funeral home of choice) following the death of a patient to offset the cost of funeral and burial services.Bereavement website: www.stjude.org/bereavement [12] provides families with grief information and resources, specifically centering grief due to the death of a child.Stay in Touch Program: Trained bereaved parent volunteers contact newly bereaved parents once a month for 4 months following the death of their child to offer condolences from one bereaved parent to another and identify concerns or needs that may need to be addressed by the Bereavement Coordinator. This program was created by bereaved parents to serve other newly bereaved parents.Resource assistance: The Bereavement Coordinator (L.C.) is available to all bereaved families and locates local bereavement supports for families as needed.

Below are five optional programs that are available for all bereaved parents; participation is voluntary, but families are made aware of the choices and may choose to engage at their discretion:Day of Remembrance Event: We host an annual two-day event, traditionally held on the medical campus, for families whose child has died within the previous 5 years. Families honor their child, interact with faculty and staff, and meet other bereaved families. The 2020 Day of Remembrance was held as a virtual event because of the COVID-19 global pandemic and virtually hosted 155 families. Through pre-recorded videos, families heard from faculty and staff; a panel of bereaved parents and grandparents; and Child Life Specialists, who modeled a memory-making activity. Families interacted via virtual small-group discussions facilitated by 39 trained bereaved parent volunteers. Parents sent in photographs, which were arranged into a pre-recorded slideshow for the celebration of life. Finally, special notecards were crafted by staff close to children of attendees and were then mailed or emailed to families.Podcast Series: Members of the QoLSC are working to develop a podcast series. Here, bereaved parents share their experiences with a series of grief-related topics to expand the diversity of bereavement care offerings we provide to bereaved families who have fewer local supports available in their communities.Virtual Parent-Led Group Discussions: Parents may join in quarterly virtual small-group topical discussions with other bereaved parents (or grandparents or adult siblings, depending on the group) run by trained, bereaved parent facilitators.Virtual Adviser Community (Legacy Voice): Bereaved caregivers may join this interactive listserv following the death of their child to provide parental input to improve institutional care. Members of Legacy Voice respond to staff-created surveys and discussion forums to tailor quality-improvement efforts.Bereaved Parent Mentoring Program: A formalized, peer-to-peer support service for bereaved parents was established to support parents with children nearing the end-of-life and/or after the death of a child. A steering council of psychosocial staff and parent mentors oversees the program. A structured process was created to match referred parents with a bereaved parent mentor. Parent mentors, who must be 2 or more years from the death of their child, are carefully vetted and receive formal training in mentorship skills, logistics of the mentor role, understanding boundaries and limitations, and mentor self-care. The role of the parent mentor is to provide a level of support distinct from that provided by clinical staff, family, or friends due to insight into the lived experience of anticipatory grief and grief after the death of a child. The mentors regularly reach out to mentees for a designated 15-month period as an additional layer of support during early bereavement. For each contact that a parent mentor has with their mentee, an encounter summary is created; at regular intervals, these encounters are evaluated and discussed with the Bereavement Coordinator. The mentors are trained to “flag” encounters of concern to assure follow up is provided by the Bereavement Coordinator or another member of the clinical team.

## 8. Strategic Planning

Involving families in the hospital’s strategic planning, evaluation, and policy making improves children’s care by enabling best practices to be shaped by families and professionals collaborating with a common goal and different perspectives. A primary goal of our institution’s strategic plan is to improve the patient and family experience, which necessitates feedback and support from caregivers.

Currently, the QoLSC have the following six goals embedded within the institution’s strategic plan for 2021 (Table 3): parents want to continue to serve as educators in healthcare communication (Recommendations #3–5). The QoLSC also plans to increase access to useful information for families at our hospital and beyond (Recommendation #4) via electronic platforms, including the bereavement website, the bereaved parents’ podcast series, and the Together.org website. In collaboration with the QoLA clinical team, QoLSC members are streamlining end-of-life care by contributing to a clinical care pathway for patients admitted to the hospital at the end of life (Recommendation #7). Finally, in partnership with the Bereavement Coordinator, bereaved parents will continue to improve the quality and diversity in bereavement supports by partnering with St. Jude affiliate partners (a network of hospitals and clinics caring for pediatric oncology patients remote from campus) to provide a continuum of bereavement support to bereaved families (Recommendation #8). 

## 9. Conclusions

Dedication to providing excellent care during treatment and into bereavement for seriously ill children and their family members is crucial for holistic, patient- and family-centered care. Parental grief is complex and support for bereaved parents is universally needed, albeit with individual differences [4,13]. Our institution has formed a strong partnership with bereaved parents as key stakeholders for program development in pediatric palliative care. Recognizing parent’s needs, an innovative institutional plan allowed bereaved parents to craft priorities for palliative care services. By leveraging motivated bereaved parents to help design, build, and staff programming that supports patients and families, healthcare providers, and students, we have improved the quality of care that we provide. Parents identified increased bereavement support as a critical need; their recommendations have guided the development of a comprehensive bereavement program within our institution [5]. Previous work with bereaved parents has shown how critical maintaining hope is across a child’s illness journey and how this hope changes as the disease trajectory evolves [14]. By engaging in this work and making these important contributions, bereaved parents make meaning of their child’s illness and death and find new hope in building their child’s legacy for years to come. This multi-faceted program will continue to develop and change under the guidance of bereaved parent partners who strive to use their grief to fuel positive changes in pediatric cancer care.

## Figures and Tables

**Figure 1 children-08-00472-f001:**
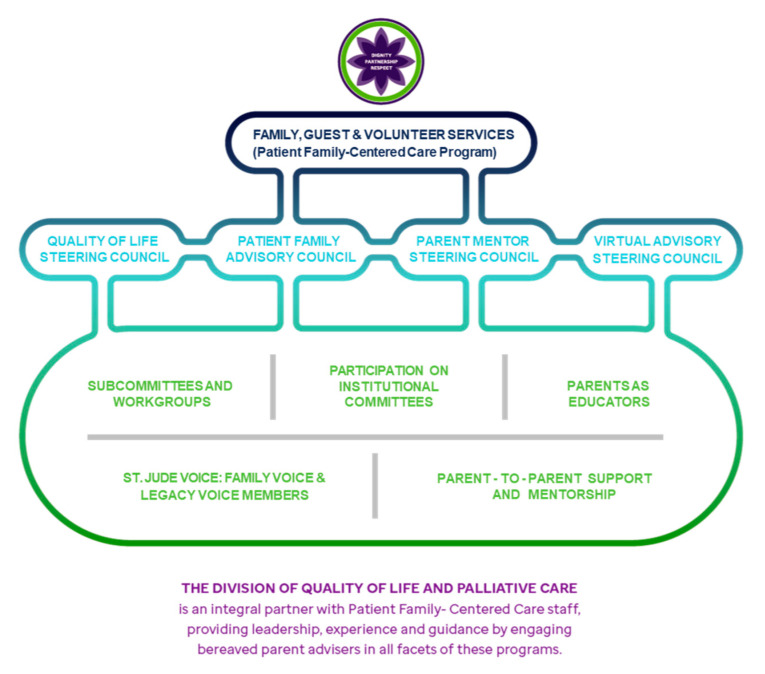
Quality of Life Steering Council (QoLSC) is part of the Patient- and Family-Centered Care Program.

**Table 1 children-08-00472-t001:** Bereaved parents’ priorities for care of children with a poor prognosis.

Eight Pivotal Parent Recommendations
Concurrent disease- and comfort-directed therapy of the highest quality Effective symptom management Relationship-based care Communication of reliable information Decision making support Care coordination Comfortable and peaceful death in the setting of their choice Bereavement care for family members and staff

**Table 2 children-08-00472-t002:** Bereaved Parents Serve as Educators.

Pediatric Subspecialty Fellow Communication Training
Half-day workshop Individual one-to-one 2 hour booster session
Institutional Clinical Staff Training (open to all healthcare providers)
End-of-Life Nursing Education Consortium (ELNEC) Quality of Life Seminar Bereavement support training
Regional Hospice Nurse Conferences
Pediatric Palliative Oncology Symposium (PPOS)
Grand Rounds and institutional didactics
Consultation with other pediatric institutions for bereavement program development

**Table 3 children-08-00472-t003:** QoLSC Strategic Planning Goals for 2021.

Quality of Life Steering Council Goals
Provide high-quality training for bereaved parent educators.
Improve quality and diversity of existing bereavement support and implement new avenues of bereavement support to fill current gaps.
Increase diversity of the Quality of Life Steering Committee.
Improve the accessibility of useful information to bereaved families everywhere.
Improve provider understanding of factors impacting patient/family/medical staff relationships.
Improve end-of-life care by collaborating with faculty and staff to standardize processes.

## Data Availability

Not applicable.

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
