# Peer review of "Leveraging Grief: Involving Bereaved Parents in Pediatric Palliative Oncology Program Planning and Development"

_children, 2021, doi:10.3390/children8060472_

Round 1

Reviewer 1 Report

While bereavement (PPC) patient, the manuscript by Spraker-Perlman has important conceptual and organizational challenges. 

  • The manuscript does not make it clear that it concerns care for a particular population, not any and all those eligible for PPC. In most reports the proportion of children who have cancer or received a stem cell transplant and are cared for by PPC programs is less than 50%.  The manuscript only briefly acknowledges that the authors’ experience relates exclusively to the world of pediatric oncology and stem cell transplantation.  The needs of patients and families where death results from congenital anomalies, extreme prematurity, neuromuscular disease, complex heart disease, etc. may have many similarities to those cared for at St. Jude’s, but this needs to be much more explicit in the manuscript, and for that matter, the title.
  • Much of the manuscript is less about the process of engaging (as announced in the title) bereaved parents and more about the what the authors (based, in part, on input from bereaved parents) believe good pediatric palliative care should look like. There is nothing wrong with that, but there is thematic confusion between the announced aim of describing “engaging” parents and a substantial proportion of the content of the paper.  These different aspects of the manuscript need to more clearly separate, perhaps even different manuscripts. The manuscript’s introduction focuses on bereavement while most of the rest of the body of the paper describes what the authors think should constitute the values and process of PPC during life.
  • While the recommendations are generally sound, if not self-evident, the discussion of each one fails to acknowledge or elaborate on clinically important nuances, such as the fact that some families perceive a conflict between establishment and maintenance of patient comfort and pursuit of cure. Sometimes that belief stems from religious views about the importance of suffering; sometimes it comes from misconceptions about role symptom management has in overall clinical management.  Another example might be when a family communicates that they do not want to hear “the truth,” especially about prognosis.  Such subtleties come up frequently enough and have sufficient importance that they deserve mention. General recommendations gloss over these crucial details facing PPC practitioners.
  • Regarding the structure of the program the authors describe, it seems odd that the QoL Steering council has only bereaved parents. Since 75-80% of childhood cancer patients survive, though many have substantial PPC needs along the way, one wonders about the apparent exclusion of parents (or older children) of survivors. The authors should describe the rationale for only including bereaved parents.
  • Again, the manuscript has two distinct components (description of good PPC and roles bereaved parents can play) that do not entirely fit with one another, at least as written.

Author Response

Response to Reviewer #1:

Thank you so much for your time, particularly during this strange 18 months as we know that additional responsibilities at home and work have made time spent reviewing papers even more precious.  We appreciate your feedback and have worked to incorporate your suggestions below, as possible in the following ways:

  1. The manuscript does not make it clear that it concerns care for a particular population, not any and all those eligible for PPC. In most reports the proportion of children who have cancer or received a stem cell transplant and are cared for by PPC programs is less than 50%.  The manuscript only briefly acknowledges that the authors’ experience relates exclusively to the world of pediatric oncology and stem cell transplantation.  The needs of patients and families where death results from congenital anomalies, extreme prematurity, neuromuscular disease, complex heart disease, etc. may have many similarities to those cared for at St. Jude’s, but this needs to be much more explicit in the manuscript, and for that matter, the title.

Thank you so much for mentioning this – ironically, the authors receive feedback when the institution and/or specific population for PPC services is mentioned (i.e. “too oncology focused”) AND also receive contrary feedback when we choose to focus on the broader PCC community. 

The authors have added commentary to assure that it is spelled out that our institutions primarily cares for patients with hematology-oncology disorders, but as we have expanded our services into ID (St. Jude cares for all children and adolescents with HIV in our catchment area) and Neurology (in particular progressive, neurological diseases like SMA, muscular dystrophies, etc.), our program is open to any and all patients who may benefit from PC services.

While our hospital is unique in that we do not serve the community in the way other academic centers may, the authors believe that many principles in this manuscript apply to a broader PPC audience.  We have noted that St. Jude is an academic hospital serving children with serious and/or life-limiting illness, in particular cancer and blood disorders in lines #34-35, 38, 53,

Here we have tried to both acknowledge the primary mission of St. Jude and assure that recommendations are applicable to non-oncology pediatric palliative care populations.

  1. Much of the manuscript is less about the process of engaging (as announced in the title) bereaved parents and more about the what the authors (based, in part, on input from bereaved parents) believe good pediatric palliative care should look like. There is nothing wrong with that, but there is thematic confusion between the announced aim of describing “engaging” parents and a substantial proportion of the content of the paper.  These different aspects of the manuscript need to more clearly separate, perhaps even different manuscripts. The manuscript’s introduction focuses on bereavement while most of the rest of the body of the paper describes what the authors think should constitute the values and process of PPC during life.

There are two distinct components to this paper – 1) the foundations of good care as defined by the parents who initiated and continue to contribute to our clinical PPC program and 2) our work towards building out these recommendations in our institutional practice.  We chose to focus on our work in bereavement as this was a major gap prior to our initial partnership with parents to define what was missing.  I disagree that this is “more about what the authors believe good palliative care should look like” as these original recommendations were drafted by parents after focus group work with these parents.  The themes that surfaced were written up by several parents and presented to our Administration.  The authors have attempted to make that clearer in lines #64-69 and lines #179-189.  Knowing that parents were engaged and driving this process from the inception, we do not feel that using the word “engaging” in the title is an oversell.  We have also tried to assure that there is a true separation of the parental priorities for the institution and bereavement programming, used as an example of how we have partnered with bereaved parents to achieve these recommendations.

  1. While the recommendations are generally sound, if not self-evident, the discussion of each one fails to acknowledge or elaborate on clinically important nuances, such as the fact that some families perceive a conflict between establishment and maintenance of patient comfort and pursuit of cure. Sometimes that belief stems from religious views about the importance of suffering; sometimes it comes from misconceptions about role symptom management has in overall clinical management.  Another example might be when a family communicates that they do not want to hear “the truth,” especially about prognosis.  Such subtleties come up frequently enough and have sufficient importance that they deserve mention. General recommendations gloss over these crucial details facing PPC practitioners.

The authors agree that there is not enough space to delineate every nuance in the practice of clinical pediatric palliative care in this paper.  We respect the points that you have delineated, but we stand behind the general guidelines we present as they were written by parents.  It is our work to support and honor them and their children with this piece, and not overly complicate that message with possible, though not exhaustive, intricacies as above.  Full papers could be written about each of the topics brought up by this reviewer.

  1. Regarding the structure of the program the authors describe, it seems odd that the QoL Steering council has only bereaved parents. Since 75-80% of childhood cancer patients survive, though many have substantial PPC needs along the way, one wonders about the apparent exclusion of parents (or older children) of survivors. The authors should describe the rationale for only including bereaved parents.

Our Patient Family Advisory Committee (PFAC) is made up of patients and families with children on- or post-therapy who are doing well in terms of disease control and bereaved parents who had children treated at our institution.  Our QoL Steering Council is a subset of our Patient Family Centered Care Program (PFCCP) that is comprised solely of bereaved parents working to improve bereavement care.  Family members from the PFAC and the QoLSC all work together on larger projects affecting the entire institution (i.e. bereaved and non-bereaved parents).  We are sorry this wasn’t clear, and we have attempted to explain how we work with all interested parents to improve care at our institution, regardless of disease type and/or prognosis in Lines #198-205. 

5. Again, the manuscript has two distinct components (description of good PPC and roles bereaved parents can play) that do not entirely fit with one another, at least as written.

As above in response to #2 (as this is the same comment), we have tried to clarify that the primary goal of this paper is to present the quality metrics defined by parents at our institution for PPC near the inception of our clinical and institutional programming.  We use the work we have done to improve bereavement support and resources as an example of one area of parental concern that has greatly improved based on the experiences of and dedication to improvement by our bereaved parents.  Hopefully these changes have satisfied the reviewers suggestions.

Reviewer 2 Report

The process described here could be very helpful to other hospitals seeking to engage parents in program development. The recommendations and description of how these recommendations turned into actual programming is valuable and inspiring information. 

Some suggestions:

  1. I wonder about the title "Productive grief." While it makes sense in the context of the paper, I hesitate to add something that might be misinterpreted as subjective judgement to grief.  
  2. I had wondered if there might be some quotes with respect to the recommendations. These recommendations are so powerful and hearing them in the parents' own words could be impactful. I realize that so many years later, this information may not be available but if it is then it could be nicely additive.
  3.  For the bereaved parents who serve on the council - why 11 parents? and do they rotate on and off? Is there diversity of parent experience - child's disease, child's age, socioeconomic, ethnicity, primary language, health literacy, other demographics. 
  4. Did the QoLA clinical team arise from the parent recommendations? It seems like the recommendations may have supported a team that was already emerging. I think more description of how exactly the parent recommendations helped the team grow would be helpful.
  5. While all 8 recommendations are important and essential, the bulk of the programming really seems to focus on bereavement support. This is such a needed area that it made me wonder if restructuring the paper focusing on recommendation #8 and the programs that have arisen to meet this need would have been stronger. 

Author Response

Response to Reviewer #2:

Thank you so much for your time, particularly during this strange 18 months as we know that additional responsibilities at home and work have made time spent reviewing papers even more precious.  We appreciate your feedback and have worked to incorporate your suggestions below, as possible in the following ways:

  1. I wonder about the title "Productive grief." While it makes sense in the context of the paper, I hesitate to add something that might be misinterpreted as subjective judgement to grief.  

We can see that this could be interpreted many ways and certainly would not want to judge anyone’s grief journey.  We have changed the title to “Leveraging Grief: Engaging Bereaved Parents in Pediatric Palliative Oncology Program Planning and Development” to hopefully portray that both a parent’s grief is important, yet also can be transformative (to not only themselves, but institutions as well).  We consistently get feedback from our bereaved parents on the QoLSC how meaningful they find their contribution to this committee and how they see their volunteered time as a direct reflection of their child’s legacy.

2. I had wondered if there might be some quotes with respect to the recommendations. These recommendations are so powerful and hearing them in the parents' own words could be impactful. I realize that so many years later, this information may not be available but if it is then it could be nicely additive.

This is an excellent recommendation, and as you said, the data is no longer new, and we no longer have much of the primary text.  It would in fact be a great addition to this paper!

3. For the bereaved parents who serve on the council - why 11 parents? and do they rotate on and off? Is there diversity of parent experience - child's disease, child's age, socioeconomic, ethnicity, primary language, health literacy, other demographics. 

We have by-laws that allow up to 12 members on the QoLSC, but we currently have 11 bereaved parent members who are serving (during the pandemic).  Each member is appointed for a 1-year term and their term can be renewed up to 3 times (to have a member serve for 3 contiguous years if the parent is in good standing (defined as attending at least 75% of activities, etc.) and would like to continue to serve).  This does allow for parents to rotate on and off the committee while encouraging new parents to be onboarded to assure that the committee contains parents of different genders, races, diagnoses/ages (of their children), health literacy levels, etc.  One weakness to date has been the inability to include non-primary English speaking parents, but we are currently working with our medical interpreters to promote participation for those that have limited English proficiency (and to define the needs of these group as they may be very different). We added some of these details to the paper in lines 221-223.

  1. Did the QoLA clinical team arise from the parent recommendations? It seems like the recommendations may have supported a team that was already emerging. I think more description of how exactly the parent recommendations helped the team grow would be helpful.

Great question!  The QOLA Team was small and under-supported at its inception around 2008.  Engaging these parents definitely helped to shine a light on the gaps in care faced by parents with children who might not survive their disease.  We have added additional details to better explain this in lines #242-248.

5. While all 8 recommendations are important and essential, the bulk of the programming really seems to focus on bereavement support. This is such a needed area that it made me wonder if restructuring the paper focusing on recommendation #8 and the programs that have arisen to meet this need would have been stronger. 

Thank you so much for these kind words.  Yes, we used Recommendation #8 as an example of what we have been able to create based on parental needs and feel (like the parents we work with) that bereavement support is both crucial and underprovided.  Some of our author group however has authored another paper focused solely on our bereavement program (Jennifer M Snaman 1Erica C Kaye 2Deena R Levine 3Brittany Cochran 4Robin Wilcox 3Charlene K Sparrow 5Nancy Noyes 6Lisa Clark 3Wendy Avery 7Justin N Baker 3 Empowering Bereaved Parents Through the Development of a Comprehensive Bereavement Program.  J Pain Symptom Manage 2017 Apr;53(4):767-775.  doi: 10.1016/j.jpainsymman.2016.10.359. Epub 2016 Dec 30. PMID: 28042068) so we had hoped to focus this paper more on our progress on all 8 recommendations.  We have made some changes to shift the focus, hopefully, more towards the parents recommendations with #8 as an example of the work we have accomplished.  

Round 2

Reviewer 1 Report

The changes the authors have made clear up the confusion created by the first version.  In the original it was not at all clear where the recommendations came from--now it is.  The specification of the special nature of the authors' institution is also helpful.  The description of the structure of the PFAC in the present version is importantly clarifying.

The authors should understand that the first review was not critical of the content, only confusion engendered by aspects of the presentation.  It continues to remain unclear to this reviewer that the word "Engaging" in the title fits with the manuscript, though that is a minor matter.  

There is a typo on line 270 of page 5 ("consultatiom").

Author Response

Thank you for this helpful input.  We changed the word "engaging" to "involving" in the title for clarity.  We also made minor corrections (typos) to this version.